# Heartbeat perception is causally linked to frontal delta oscillations

David Haslacher[1,6], Philipp Reber [1,2,6], Alessia Cavallo[1], Annika Rosenthal[1], Elisabeth Pangratz[1], Anne Beck[3], Nina Romanczuk-Seiferth[4], Vadim Nikulin[5], Arno Villringer [5] & Surjo R. Soekadar [1] ✉

The ability to accurately perceive one's own bodily signals, such as the heartbeat, plays a vital role in physical and mental health. However, the neurophysiological mechanisms underlying this ability, termed interoception, are not fully understood. Converging evidence suggests that cardiac rhythms are linked to frontal brain activity, particularly oscillations in the delta (0.5 – 4 Hz) band, but their causal relationship remained elusive. Here, we identified a frontal network of delta oscillations that was anticorrelated with both heartbeat perception and heartbeat-evoked brain responses. Using amplitude-modulated transcranial alternating current stimulation (AM-tACS), a method to enhance or suppress brain oscillations in a phase-specific manner, we investigated whether frontal delta oscillations are causally linked to heartbeat perception. We found that enhancement of delta phase synchrony suppressed heartbeat detection accuracy, while suppression of delta phase synchrony enhanced heartbeat detection accuracy. These findings suggest that frontal delta oscillations play a significant role in heartbeat perception, paving the way for causal investigations of interoception and potential clinical applications.

Interoception refers to the sense by which individuals perceive the physiological state of their body, such as satiation, hydration, respiration, and the heartbeat[1]. It was found that accurate interoception is essential for the effective regulation of physiological processes, ensuring that the body's response to environmental challenges is appropriate. For instance, accurate perception of hydration leads to appropriate fluid intake, which in turn maintains body function[2]. Moreover, the integrity of interoception ensures that cardiovascular, metabolic, and other physiological states can be adjusted to the individual's current and future requirements[3,4]. In turn, dysfunctions in interoception were found in a range of mental health conditions, including anxiety[5], depression[6], panic disorder[7], eating disorders[8], and substance use disorders[9].

Similar to sensorimotor processes, interoception forms a perception-action loop[10]. Interoceptive signals, originating from sensory receptors within internal organs, reflect information about physiological states that is integrated with other information in the insular and anterior cingulate cortices[1,4,11]. For instance, the perception and integration of afferent cardiac signals has been shown to be reflected in neural responses that are time-locked to the cardiac cycle, so-called heartbeat-evoked potentials (HEPs)[12]. Efferent signals are then generated in the form of behavioral responses (e.g., eating or drinking) or autonomic adjustments (e.g., increasing heart rate).

This bidirectional interplay ensures an adaptive response to both perturbations of the body state and environmental demands[10,13].

It was found that the interaction between brain and body is intrinsically rhythmic, spanning over a wide range of frequencies[14]. For instance, the gastric basal rhythm oscillates at about 0.05 Hz, respiration at about 0.25 Hz, and the heartbeat at about 1 Hz. Although the link between these rhythms and those of the brain is unclear, there is increasing evidence of both afferent and efferent forms of temporal locking in the sense of *entrainment*[15–20]. Moreover, several studies found various links between frontal delta oscillations (FDOs, 0.5–4 Hz) and autonomic functions[21], such as an anticorrelation between frontal delta power and the heart rate[19]. At the same time, it was found that cardiac activity can modulate FDOs[20,22]. These results suggest that FDOs play an important role in the bidirectional coupling between the brain and the heart. However, it is not clear how this coupling relates to heartbeat perception. Since brain responses related to self-generated sensory input are typically suppressed[23–25], we hypothesized that phase synchrony of FDOs is causally and negatively linked to heartbeat-evoked potentials and heartbeat detection.

To test this hypothesis, we invited healthy human volunteers ($N = 24$) to perform an established heartbeat detection task[11] while electroencephalography (EEG) was recorded (Fig. 1). In this task, participants had

[1]Department of Psychiatry and Neurosciences, Charité – Universitätsmedizin Berlin, Berlin, Germany. [2]Department of Psychology, University of California, Berkeley, California, USA. [3]Institute for Mental Health and Behavioral Medicine, Department of Psychology, HMU Health and Medical University, Potsdam, Germany. [4]Department of Psychology, MSB Medical School Berlin, Berlin, Germany. [5]Department of Neurology, Max Planck Institute for Human Cognitive and Brain Sciences, Leipzig, Germany. [6]These authors contributed equally: David Haslacher, Philipp Reber. ✉e-mail: surjo.soekadar@charite.de

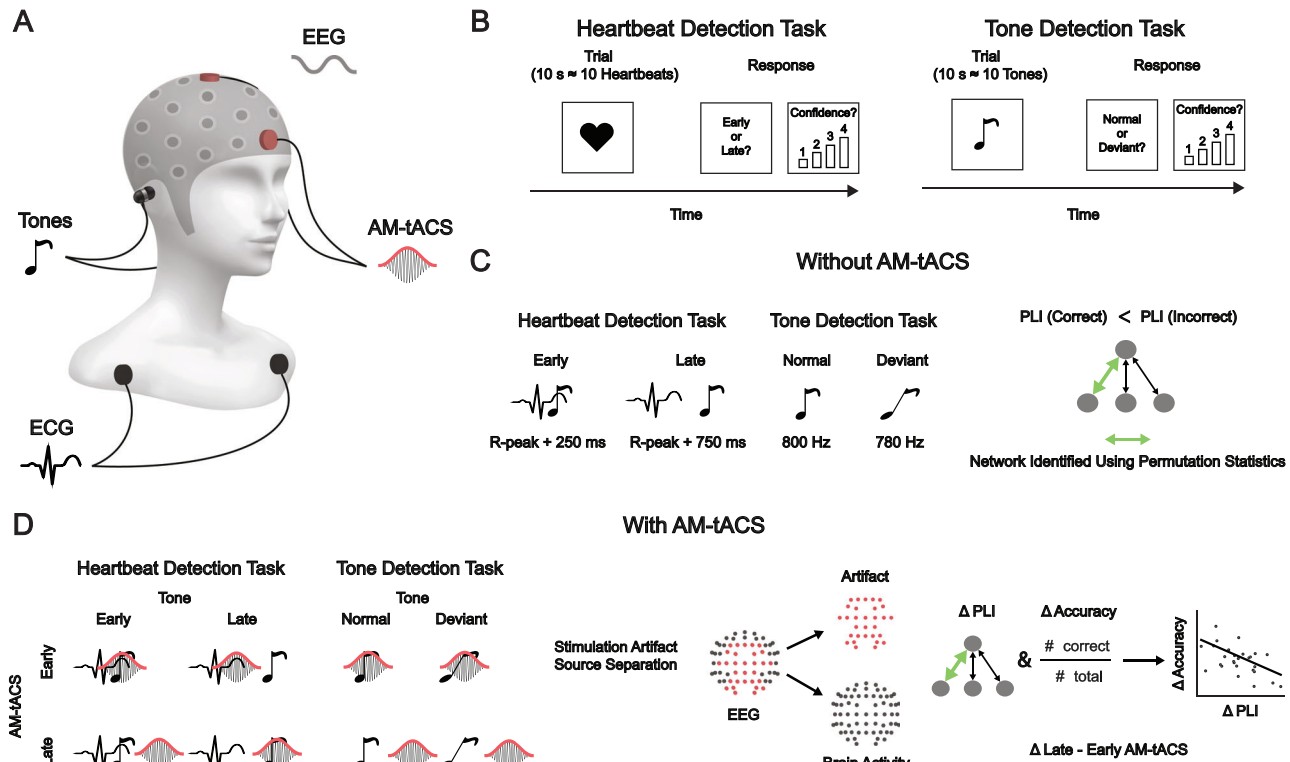

**Fig. 1 | Experimental paradigm. A** Electroencephalography (EEG) and electro-cardiography (ECG) were recorded while auditory stimuli were delivered through a pair of earphones. To modulate delta phase synchrony, amplitude-modulated transcranial alternating current stimulation (AM-tACS) was applied to the frontal cortex. Parameters of AM-tACS comprised an 8 kHz carrier frequency, an envelope frequency corresponding to the heart rate, and a stimulation intensity individually adjusted to avoid any somatosensory perception (6.28 ± 0.970 mA). **B** During the heartbeat detection task[11], participants (*N* = 24) were asked to assess whether their heartbeat was early or late relative to the sequence of pure tones. During the tone detection task, participants were asked to assess whether the sequence of pure tones contained a deviant tone. Auditory stimulation and trial length (10 heartbeats) were identical across tasks, such that the only difference consisted of the focus of attention, and trials of each task were mixed such that they were performed in a pseudorandomized order. **C** In absence of AM-tACS, pure tones were played either early with the perceived heartbeat, (~250 ms after ECG R-peak) or late (~750 ms after ECG R-peak). Auditory stimuli consisted of a sequence of normal (800 Hz) tones, in some trials containing a deviant (785 Hz) tone. To identify the network of delta oscillations linked to heartbeat perception, a network-based permutation test was used to assess where the phase lag index (PLI) was higher during incorrect than during correct responses in the heartbeat detection task. **D** Participants performed the same tasks in the presence of AM-tACS, which was applied early or late relative to the heartbeat. We show only one cycle of AM-tACS for illustrative purposes, although it was applied (and adjusted to the heartbeat) continuously. Stimulation artifact source separation (SASS) was used for stimulation artifact removal from EEG signals. Finally, modulation of delta phase synchrony in the network previously identified was correlated with the modulation of heartbeat detection accuracy.

to indicate whether an auditory tone sequence was presented early or late relative to the heartbeat. Performance in this task thus measures heartbeat detection, a form of interoceptive accuracy[26]. To differentiate EEG signals related to auditory processing, participants also performed an auditory tone detection task in which they were asked to indicate deviant tones within a tone sequence.

Assessing the entrainment of brain oscillations by interoceptive signals such as the heartbeat comes with methodological challenges. By aligning their high- or low-excitability phases with rhythmic input, brain oscillations constitute an efficient mechanism for filtering of predictable sensory signals[27,28]. Typically, this phenomenon is studied by comparing the phase of brain oscillations at the time of enhanced and suppressed sensory input[28,29]. However, due to heartbeat-locked artifacts in the EEG[30–32], the phase of delta oscillations relative to the heartbeat is difficult to assess directly. Instead, since coordinated entrainment of brain oscillations manifests in network synchrony[33], we chose to assess delta phase synchrony between brain regions by computing the phase lag index (PLI) to mitigate the influence of cardiac artifacts[34]. Subsequently, we applied amplitude-modulated transcranial alternating current stimulation (AM-tACS), a frequency-tuned form of non-invasive brain stimulation to enhance or suppress brain oscillations[35,36], to the frontal cortex either early or late relative to the heartbeat. We expected that AM-tACS results in phase-dependent enhancement and suppression of delta phase synchrony in the identified network, causing an increase and decrease in heartbeat detection accuracy.

## Results

### Frontal delta phase synchrony is associated with heartbeat perception

We found that delta phase synchrony was associated with heartbeat detection accuracy in a network of frontal brain regions (Fig. 2A), which exhibited higher PLI during incorrect than during correct responses (t(23) = 1.61, p = 0.044, $d_z$ = 0.336, 95% CI [0.012, 0.641], network-based permutation test), confirming that delta phase synchrony was negatively associated with heartbeat detection accuracy. We also assessed whether PLI in the identified network was larger during the tone detection task than during the heartbeat detection task (Fig. 2B). No such difference was found (t(23) = −0.218, p = 0.623, $d_z$ = −0.0454, 95% CI [−0.361, 0.272], permutation test), demonstrating that increased delta phase synchrony was associated with decreased heartbeat perception but not tone perception. Correspondingly, a two-way ANOVA revealed an interaction between task (heartbeat or tone detection) and response (correct or incorrect) with respect to frontal delta synchrony (F(1,23) = 5.90, p = 0.0233). We also assessed whether differences in oscillatory synchrony between incorrect and correct responses were restricted to the delta band by performing the network-based permutation test for the

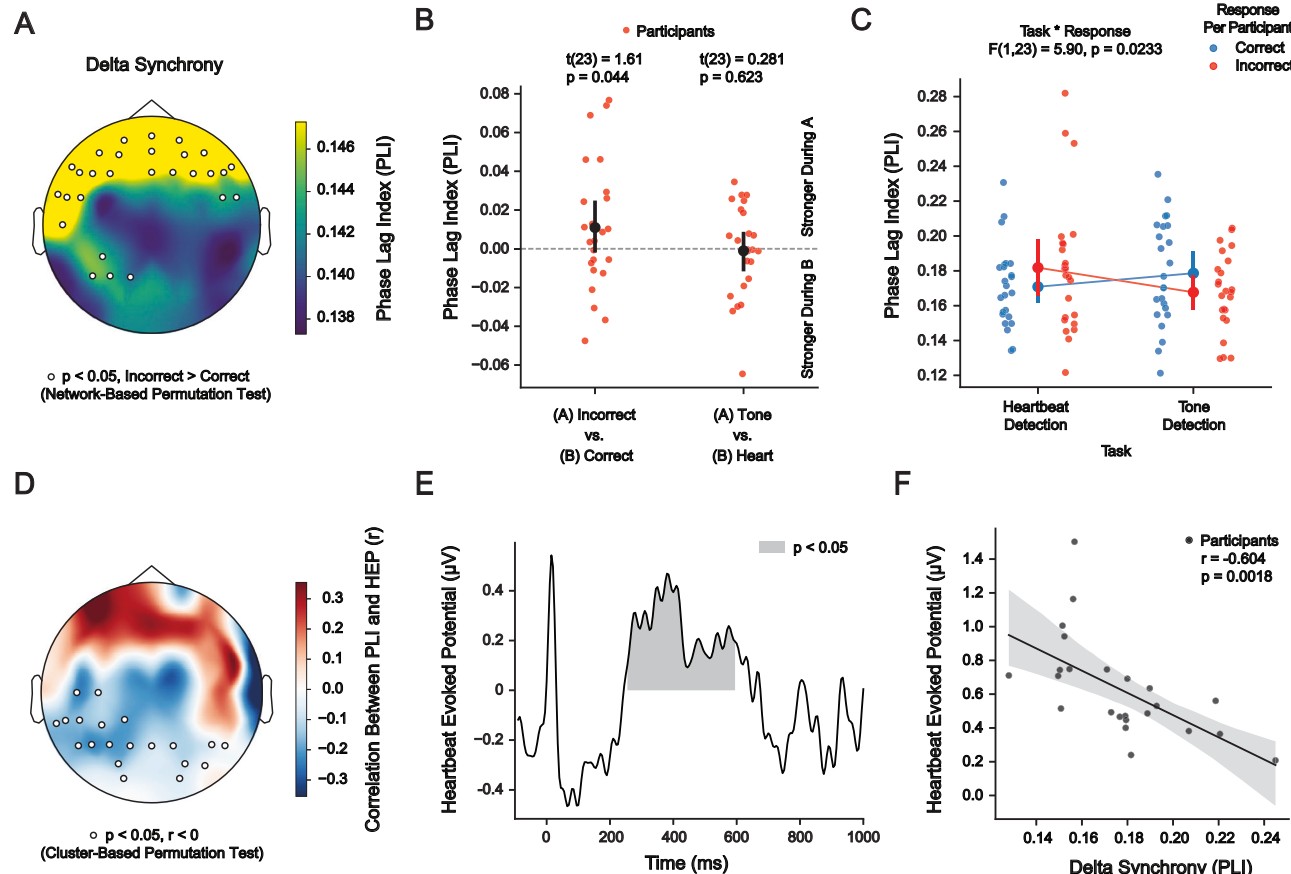

**Fig. 2 | Increased frontal delta phase synchrony is associated with reduced heartbeat perception and reduced heartbeat-evoked potential. A** Phase synchrony of delta oscillations in a network recorded over frontal brain areas was associated with heartbeat detection. **B** Delta phase synchrony (averaged within the identified network) was larger during incorrect than during correct responses in the heartbeat detection task. No difference in delta phase synchrony between the heartbeat and tone detection tasks was found. **C** An interaction between task (heartbeat or tone detection) and response (correct or incorrect) was found with respect to frontal delta synchrony. **D** Delta phase synchrony (averaged within the identified network) was anticorrelated with the amplitude of the heartbeat-evoked potential (HEP) in several clusters of central, parietal, and occipital EEG electrodes. **E** The anticorrelation between delta phase synchrony and HEP amplitude was found between 255 and 595 ms after the ECG R-peak. **F** Delta phase synchrony (averaged within the identified network) was strongly anticorrelated with the HEP amplitude (averaged over significant electrodes and timepoints). Error bars throughout this figure represent the 95% confidence interval.

theta (4–8 Hz), alpha (8–15 Hz), beta (15–30 Hz), and gamma (30–45 Hz) bands. No differences in synchrony between incorrect and correct responses were found in these frequency bands. Cluster-based permutation testing did not reveal any difference in delta amplitude between correct and incorrect responses.

We next assessed whether PLI in the identified network was associated with HEP amplitude in any brain region. We found an anticorrelation between PLI (calculated over the entire trial duration) and HEP amplitude in central, parietal, and occipital areas from 255 to 595 ms after the ECG R-peak ($p < 0.05$, cluster-based permutation test) (Fig. 2D and 2E). The strength of this anticorrelation strongly increased when HEP amplitudes were averaged across significant sensors and timepoints ($r = -0.604$, $p = 0.0018$) (Fig. 2F). To further confirm that delta phase synchrony was associated with heartbeat perception and not tone perception, we assessed whether PLI in the identified network was associated with auditory evoked potential (AEP) amplitude. Cluster-based permutation testing did not reveal any correlation between PLI and AEP amplitude.

We also assessed whether evoked responses or physiological artifacts in the delta frequency range activity might explain our findings (Fig. S1). In addition to endogenous frontal delta oscillations, heartbeat-evoked activity, as well as auditory-evoked activity, were observable during our task (Fig. S1A and S1B). Additionally, cardiac and ocular artifacts may be mistaken for delta-band brain activity in the recorded signal[37]. We found that frontal delta synchrony, particularly its difference between incorrect and correct responses, was not accounted for by evoked responses. Neither was it accounted for by ocular or cardiac artifacts (Fig. S1).

## Transcranial alternating current stimulation modulates frontal delta phase synchrony and heartbeat perception

We first assessed whether the timing of AM-tACS relative to the ECG R-peak influenced heartbeat perception (Fig. 3A). A one-tailed test revealed that heartbeat detection accuracy during late AM-tACS (66.0 ± 14.7%) was lower ($t(24) = -2.06$, $p = 0.0256$, $d_z = -0.429$, 95% CI [−0.774, −0.050]) than during early AM-tACS (69.2 ± 15.7%). Likewise, signal discriminability during late AM-tACS (0.855 ± 0.818 d') was lower ($t(24) = -2.18$, $p = 0.040$, $d_z = -0.454$, 95% CI [-0.887, −0.021]) than during early AM-tACS (1.11 ± 0.981 d'). The response bias during late AM-tACS (-0.183 ± 0.558 c) was comparable to ($t(24) = -0.991$, $p = 0.332$, $d_z = -0.207$, 95% CI [−0.624, 0.210]) the response bias during early AM-tACS (-0.111 ± 0.658 c). Before assessing delta oscillations recorded in the presence of AM-tACS, we confirmed that electric stimulation artifacts were successfully attenuated by SASS (Figs. S2 and S6). We then confirmed that AM-tACS enhanced and suppressed frontal delta phase synchrony in a phase-dependent manner ($p_{group} = 3.23 \times 10^{-10}$, Fig. S4). We found that the change in delta phase synchrony across delay conditions predicted ($r = -0.411$, $p = 0.0231$) the change in heartbeat detection accuracy (Fig. 3B). Thus, an increase in delta phase synchrony led to a decrease in heartbeat detection accuracy, and vice-versa. Cluster-based permutation

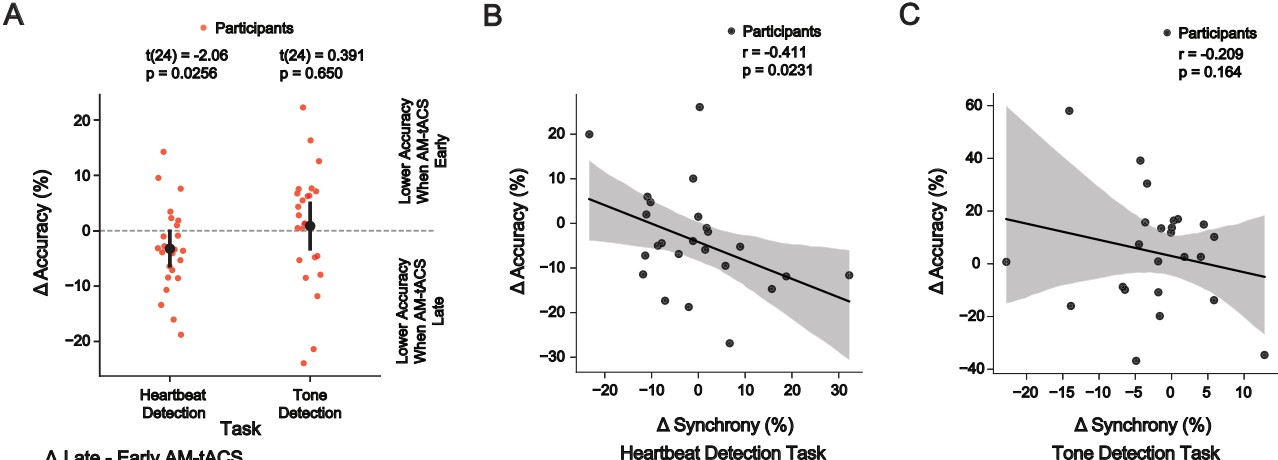

**Fig. 3 | Amplitude-modulated transcranial alternating current stimulation (AM-tACS) suppresses heartbeat detection accuracy by enhancing frontal delta phase synchrony. A** When AM-tACS was applied late relative to the heartbeat, heartbeat detection accuracy was lower than when AM-tACS was applied early relative to the heartbeat. No differential effect on tone detection accuracy was found. **B** The modulation of delta phase synchrony (difference between late and early AM-tACS conditions) in the target network (identified in Fig. 2A) was anticorrelated with the modulation of heartbeat detection accuracy, such that an enhancement of delta phase synchrony caused a suppression of heartbeat detection accuracy. **C** No relationship between the modulation of delta phase synchrony and tone detection accuracy was found. Error bars throughout this figure represent the 95% confidence interval.

testing did not reveal any difference in delta amplitude between AM-tACS delay conditions.

To exclude that the physiological and behavioral effects of AM-tACS are related to neural processing of the auditory stimulus, we performed the same analyses for the tone detection task. We found that tone detection accuracy during late AM-tACS ($49.7 \pm 5.51\%$) was comparable to ($t(24) = 0.391$, $p = 0.650$, $d_z = 0.0816$, 95% CI [$-0.2841, 0.4405$]) tone detection accuracy during early AM-tACS ($48.9 \pm 7.11\%$). Likewise, signal discriminability during late AM-tACS ($-0.00252 \pm 0.298$ d') was comparable to ($t(24) = 0.347$, $p = 0.731$, $d_z = 0.0724$, 95% CI [$-0.341, 0.485$]) signal discriminability during early AM-tACS (-0.0425 ± 0.358 d'). Response bias during late AM-tACS ($-0.225 \pm 0.273$ c) was comparable to ($t(24) = -1.50$, $p = 0.147$, $d_z = -0.313$, 95% CI [$-0.736, 0.110$]) response bias during early AM-tACS ($-0.151 \pm 0.251$ c). No relationship ($r = -0.209$, $p = 0.164$) between the change in synchrony and the change in accuracy across delay conditions was found (Fig. 3C).

While we find a link between changes in frontal delta synchrony and changes in heartbeat detection accuracy due to AM-tACS (Fig. 3B), we did not find a difference in frontal delta synchrony between late and early AM-tACS conditions. We reasoned that this might result from a variability in the optimal phase difference between AM-tACS and frontal delta oscillations to enhance (or suppress) frontal delta synchrony. Indeed, we find that this optimal phase difference varies across participants (Fig. S4). We also find that this phase difference varies across participants within both the early and late AM-tACS conditions (Fig. S3).

## Discussion
### Interoceptive predictive coding
Our findings suggest that FDOs may suppress heartbeat-evoked brain activity, attenuating perception of the heartbeat. While previous studies have found that processing of visual[18,38–40], auditory[41], and somatosensory[42–45] stimuli depends on the timing of stimulus presentation relative to the heartbeat, the underlying mechanism remained unknown. Our findings suggest that such heart cycle-dependent modulations of perception may be mediated by FDOs, consistent with the notion that they might reflect unconscious predictions of bodily signals such as the heartbeat[44,46–48].

In predictive coding, feedback predictions inhibit the feedforward processing of predicted stimuli, such that only the unpredicted sensory information is propagated upwards through the cortical hierarchy[47,49]. Prior

empirical and conceptual work has converged on a model in which low-frequency (<30 Hz) oscillations in deep cortical layers convey predictions, while high-frequency (> 30 Hz) oscillations in superficial cortical layers convey prediction errors[47,50–52]. Thus, increased frontal delta synchrony during missed heartbeats (Fig. 2B) may reflect interoceptive predictions of the heartbeat emanating from agranular visceromotor cortices such as the anterior cingulate and ventromedial prefrontal cortices[47]. These predictions would be propagated downwards to primary sensory areas such as the somatosensory cortex, where they inhibit feedforward processing, resulting in an attenuated HEP (Fig. 2F). Thus, our findings are consistent with prior work linking the HEP to heartbeat prediction errors[53]. In predictive coding, attention implements the precision-weighting of predictions and prediction errors by modulating the gain of feedforward and feedback processing[49]. Correspondingly, prior work has found that attention modulates the amplitude of the heartbeat-evoked potential[54]. In our work, however, we do not find a difference in frontal delta synchrony between heartbeat and tone detection tasks (Fig. 2B). This may be explained by evidence that the medial prefrontal cortices implement supramodal prediction mechanisms that extend beyond interoception to include exteroceptive signals, such as those involved in auditory perception[55].

Importantly, our findings suggest that (interoceptive) predictive coding is amenable to experimental investigation using phase-specific brain stimulation. Future studies should also investigate whether our findings can be generalized to the processing of other interoceptive signals beyond heartbeat[21], and how manipulation of these interoceptive signals affects FDOs[56]. An important methodological difference between our study and prior work[55] investigating the role of brain oscillations in mediating the perception of rhythmic sensory input should also be emphasized. We did not investigate an alignment (synchronization) of delta oscillations with the heartbeat, which was difficult in the presence of cardiac-related artifacts in the EEG. To mitigate this issue, we investigated synchronization of delta oscillations within a prefrontal brain network using the PLI, a measure of synchronization robust to volume conduction originating from sources such as the cardiac artifact.

### Interoception, emotion, and clinical applications
Causal modulation of FDOs via AM-tACS offers an approach to verify their functional relevance for interoception and could be applied to experimentally test basic theories of emotion, which make competing predictions about the causal role of interoception. Growing evidence underscores the

importance of interoception for emotional and motivational processes[1,46,47,57]. Classical theories, such as the James-Lange theory, posit that emotions are a response to bodily signals[58]. Conversely, the Cannon-Bard theory argues that bodily signals and emotional experiences occur simultaneously but independently[59]. The Schachter-Singer theory, meanwhile, integrates bodily signals with cognitive appraisal to explain emotional experience[60]. Although these theories have historical importance, they have been extensively reviewed and critiqued over the years. Recent advancements have led to more nuanced frameworks. For instance, LeDoux's dual circuit theory emphasizes the interplay between fast, subcortical circuits and slower, cortical circuits in generating emotional responses[61]. By selectively manipulating the neural processing of bodily signals through AM-tACS, it might be possible to experimentally test not only classical but also contemporary models of emotion. Such experiments could elucidate the relative contributions of interoceptive signals to emotional processes across different theoretical frameworks, thus advancing our understanding of the neural and physiological bases of emotion.

Our results suggest that AM-tACS could be used to better understand whether alterations in interoception are causally related to symptoms of a range of health conditions. For instance, it has been proposed that altered interoceptive processing is a primary cause of anxiety and depression[5,6], in addition to other psychiatric disorders[62]. Dysfunctions of interoception have also been linked to a range of other brain disorders with somatic symptoms such as chronic pain[63], obesity[64], and chronic stress[65]. Beyond disentangling the causal relationship between interoception and other aspects of these disorders, our approach might facilitate new avenues of treatment.

## Limitations of transcranial alternating current stimulation approach

AM-tACS is a recent neuromodulation technique that was originally introduced to avoid electric stimulation artifacts of tACS at the target frequency in simultaneously recorded M-/EEG[66–68]. Thus, AM-tACS is primarily used to assess the effects of tACS during stimulation[35,36,66]. Here, we used AM-tACS to assess enhancement and suppression of frontal delta synchrony depending on the phase difference between AM-tACS and frontal delta oscillations. However, it should be noted that the use of AM-tACS is associated with several challenges. First, residual stimulation artifacts at the target frequency caused by non-linear characteristics of the stimulation and recording hardware may confound observed results[68]. To mitigate this possibility, we have applied SASS, an artifact suppression technique specifically designed for use with AM-tACS[67], and confirmed that frontal delta synchrony after application of SASS is comparable to that in the absence of AM-tACS (Fig. S2). We also provide a similar analysis for the EEG power spectral density and delta power (Fig. S6). We have also correlated the physiological and behavioral effects of AM-tACS to strengthen the interpretation that the physiological effects are not merely residual stimulation artifacts (Fig. 3). A further limitation of AM-tACS relates to its lower efficacy compared to conventional tACS. Due to the use of an amplitude-modulated stimulation waveform in AM-tACS, neurons must demodulate the envelope to exhibit a physiological response at the target frequency. Correspondingly, simulations indicate that a higher stimulation intensity is required to achieve effects of AM-tACS comparable to conventional tACS[69]. It should be noted that these simulations have not been confirmed by experimental studies. However, recent work on temporal interference stimulation (TIS), which employs a stimulation waveform equivalent to AM-tACS, has found effects at conventional intensities of 1–3 mA[70,71]. Here, we have used an individually adjusted stimulation intensity (6.28 ± 0.970 mA) substantially higher than conventional stimulation intensities (approx. 1 mA).

Several methodological limitations of tACS should be addressed to reduce variability of stimulation effects across scientific and clinical applications[72–75]. Effect variability may result from inter-individual anatomical differences, which should be considered in the design of personalized stimulation montages[74]. Importantly, the effects of tACS also depend on the

brain state during stimulation[76–78]. We found that AM-tACS enhanced and suppressed FDOs in a phase-dependent manner. While this mechanism has been documented elsewhere for other brain oscillations[35,79–82], its utilization in clinical practice remains a challenge. To selectively enhance or suppress a brain oscillation using this mechanism, electric stimulation artifacts in simultaneously recorded EEG must be sufficiently attenuated to extract single-trial phase information and continually adapt tACS to the oscillation in real-time[35,36,67,83]. Furthermore, the phase difference between tACS and the oscillation must be personalized to achieve the desired enhancement or suppression effect[35,36,80,82,84]. Finally, poor focality and depth of transcranial electric stimulation[73,85,86] limit the capacity of tACS to selectively modulate subregions of the frontal brain mediating interoception. Recently developed techniques employing interfering electric[70,71,87] or magnetic[88] fields may alleviate these issues while remaining compatible with the stimulation artifact rejection approaches employed here[35,67,83]. These technical advances may enable more refined investigations of the causal link between FDOs, interoception, and brain (dys)function.

Finally, sensory co-stimulation should be assessed as a potential mechanism of tACS effects. It is known that tACS can stimulate the retina[89] and peripheral nerves[90], leading to unintended effects on brain activity and behavior. Here, we have used AM-tACS with a very high carrier frequency (8 kHz), which is known to substantially reduce sensory co-stimulation effects[91]. Furthermore, we have individually adjusted the stimulation intensity to remain below the threshold for sensory perception. Nevertheless, our study did not feature an active control condition, where the same stimulation waveform is applied to a different region of the scalp (or body), which would have been necessary to definitively exclude the contribution of sensory pathways in our observed stimulation effects[92]. Finally, we did not employ questionnaires to assess subjective quality and intensity of sensory effects[93].

## Limitations of the heartbeat detection paradigm

The heartbeat discrimination paradigm used here[11] presents several conceptual and psychometric limitations. Conceptually, the task assumes that accurate detection of heartbeat-synchronous stimuli reflects pure interoceptive ability, yet it also requires complex multimodal integration of internal (cardiac) and external (auditory) signals, making it difficult to isolate interoceptive accuracy from general perceptual or attentional processes[26]. Psychometrically, the paradigm suffers from low sensitivity and reliability; most participants perform near chance, and small variations in task parameters (e.g., stimulus delay) can substantially affect performance[94]. Moreover, standard signal detection indices (e.g., d′) in these tasks often fail to consistently identify true heartbeat detectors[26,94].

Nevertheless, despite their limitations, heartbeat discrimination paradigms remain one of the most accepted tools for assessing interoceptive accuracy, as alternative approaches such as heartbeat counting tasks suffer from additional conceptual and psychometric issues. Specifically, heartbeat counting is heavily influenced by individuals' prior beliefs about heart rate and general counting or estimation abilities, rather than reflecting genuine detection of visceral signals[26,94]. Meta-analytic findings confirm that performance in heartbeat counting correlates with performance in visual counting and does not improve with repetition, further undermining its validity as a measure of interoceptive accuracy[94]. Crucially, meta-analyses find that performance in heartbeat counting tasks does not correlate with mental health outcomes[95]. Unfortunately, the relationship between interoceptive accuracy and mental health outcomes has rarely been studied with other tasks, such that the clinical relevance of the heartbeat discrimination task used here remains to be investigated.

## Methods
### Participants
In total, 25 participants (14 female, 11 male, 26 ± 5 years of age) were invited to participate in the study and provided written informed consent in accordance with the ethics committee of the Charité – Universitätsmedizin Berlin (EA1/077/18). All ethical regulations relevant to human research

participants were followed. One participant was excluded due to exceedingly high heartbeat detection accuracy (>90%) in all conditions.

## Electroencephalography

Electroencephalography (EEG) was recorded from 64 Ag/AgCl electrodes distributed over the scalp according to the international 10–20 system using a NeurOne system (Bittium Corp., Oulu, Finland). The electrode at position C5 was used as a reference. For all recordings, the amplifier was set to DC-mode with a dynamic range of +/−430 mV, a resolution of 51 nV/bit, and a range of 24 bit. Data were sampled at 2 kHz. The impedance for all sensors was kept below 10 kΩ. Otherwise, the sensors were marked for interpolation.

## Electrocardiography

Electrocardiography (ECG) was recorded from an auxiliary bipolar channel along with EEG data. Electrodes were placed under the left and right infraclavicular fossa. The EEG system sent the ECG data to a real-time computer via a real-time UDP stream, where it was further processed to control AM-tACS and auditory stimuli.

## Transcranial alternating current stimulation

Amplitude-modulated transcranial alternating current stimulation (AM-tACS) was delivered to the scalp with a current of 6.28 ± 0.970 mA and a carrier signal frequency of 8 kHz using a Digitimer DS5 (Digitimer Ltd, UK). The amplitude of AM-tACS was adjusted individually for each participant to the maximal level that remained below the threshold for somatosensory perception. Two circular rubber electrodes (34 mm diameter, 2 mm thickness) were used with conductive ten20 paste (Weaver & Co, Aurora, CO, USA) to apply AM-tACS to the scalp. Rubber electrodes delivering AM-tACS were centered on positions Fpz and Cz of the international 10-20 system. The electric stimulator was controlled by an SDG 2042X signal generator (Siglent, NL), which applied amplitude-modulation to the carrier signal depending on an input voltage signal received from the Speedgoat Performance Real-Time Target Machine (Speedgoat GmbH, CH). The real-time target machine adjusted the phase of the amplitude envelope to the ECG signal in real-time, such that its maximum was either early (~250 ms after ECG R-peak) or late (~750 ms after ECG R-peak) relative to the heartbeat. Both frequency and phase of the AM-tACS envelope were continually adjusted to the individual heartbeat. After each ECG R-peak, the frequency was adjusted to match the average heart rate over the last 10 heartbeats, while the phase was adjusted to achieve the desired early or late timing. This timing was chosen to match that of auditory stimulation, which was chosen according to a prior study (see methods section "Auditory stimuli").

## Auditory stimuli

Auditory stimulation was the same for all experimental conditions and consisted of a sequence of pure tones of 100 ms duration delivered through earphones at a 60 dB level. In 50% of trials, the tones were played early relative to the perceived heartbeat (~250 ms after ECG R-peak). In the remaining 50%, the tones were played late relative to the perceived heartbeat (~750 ms after ECG R-peak). Additionally, in 50% of the trials, the tone sequence contained a deviant tone at a random position. The timing of early and late tones was chosen to match that of a prior study implementing the same paradigm, but with pulse measurements from the finger[11]. In that study, early tones were played immediately after the pulse wave, whereas late tones were played ~500 ms after the pulse wave. Our timing was subsequently chosen based on evidence that the pulse wave arrives at the finger ~250 ms after the ECG R-peak[96]. Tone frequency was also chosen to match the same prior study: normal tones had a frequency of 800 Hz, whereas deviant tones had a frequency of 785 Hz.

## Behavioral tasks

Participants carried out randomized trials of a heartbeat detection task and a tone detection task. For sensory stimulation parameters, see the previous section. Throughout each 10-second trial of either task, they were presented with a series of tones, which were triggered by their own heartbeat (see methods section "Auditory stimuli"). In the heartbeat detection task, we manipulated attention by presenting study participants with a heart symbol in the middle of the screen. At the end of each trial, they were asked to indicate whether the tones were early or late relative to the heartbeat, and how confident they were about their response (by selecting 1,2,3, or 4). Throughout the tone detection task, we directed them to attend to the pitch of the tones by showing them a musical note symbol in the middle of the screen. At the end of each trial, they were asked to indicate whether the tone sequence contained a deviant tone, and how confident they were about their response (by selecting 1,2,3, or 4). To compute heartbeat and tone detection accuracy, responses were weighted by their confidence.

## Stimulation artifact source separation

Following[67], we removed the AM-tACS artifact from bandpass-filtered (0.5 – 4 Hz) EEG data using SASS. First, we computed the sensor covariance matrix **A** in the presence of AM-tACS, as well as the sensor covariance matrix **B** in the absence of AM-tACS. Subsequently, a source separation matrix W was computed by joint diagonalization of **A** and **B**: W =

$$\begin{pmatrix} \mathbf{w}_1^T \\ \vdots \\ \mathbf{w}_n^T \end{pmatrix}$$ where $\mathbf{A}\mathbf{w}_i = \lambda_i \mathbf{B}\mathbf{w}_I$ and $\lambda_i = \frac{\mathbf{w}_i^T \mathbf{A} \mathbf{w}_i}{\mathbf{w}_i^T \mathbf{B} \mathbf{w}_i}$. The $\mathbf{w}_i$ were ordered from

greatest to least $\lambda I$, which represented the ratio of artifact power to brain signal power (noise-to-signal ratio) in each component. Finally, the matrix P was constructed as $\mathbf{P} = \mathbf{W}^{-1}\mathbf{S}\mathbf{W}$, where $\mathbf{S} = \begin{pmatrix} 0 & & \\ & \ddots & \\ & & 1 \end{pmatrix}$, with zeros

on the diagonal representing artifact components that were removed from the data. To remove AM-tACS artifacts, the sensor-space data were then multiplied by P. To select the number of components to reject (zeros in S), $\left| \mathbf{B} - \mathbf{P}\mathbf{A}\mathbf{P}^T \right|_2$ was minimized, resulting in 20.0 ± 11.1 rejected components.

## Electroencephalography data processing

To compute the HEP in the absence of AM-tACS, EEG data were filtered from 0.5 - 30 Hz using a finite impulse response filter. Epochs were then extracted around the ECG R-peak (-100 to 1000 ms), and the signal was averaged over epochs. To compute the delta phase, EEG data were filtered from 0.5–4 Hz using a finite impulse response filter of order 1321 (corresponding to 6.61 seconds). For data recorded in the presence of AM-tACS, SASS was then applied to suppress the electric stimulation artifact (see previous section). Subsequently, the Hilbert transform was applied to the data, and the angle of the analytic signal was taken to obtain the phase of delta activity. To assess long-range delta phase synchrony, the phase lag index was computed between each pair of channels[34], for each trial (10 heartbeats). The PLI was chosen because it mitigates the effect of volume conduction. By ignoring 0° and 180° phase differences between two sources, any interference due to volume conduction that would bias conventional measures of synchronization (like the phase locking value, PLV) is suppressed. This makes the PLI measure robust to electric stimulation artifacts and volume conduction of neurogenic sources. MNE-Python[97], NumPy[98], and SciPy[99] were used for all analyses. To remove delta-band activity linked to heartbeat-evoked potentials (HEPs), auditory-evoked potentials (AEPs), and cardiac artifacts, a template subtraction procedure was employed (Fig. S1). This template subtraction procedure was previously used to remove cardiac artifacts[100,101] and evoked responses[102] from EEG data. First, evoked responses and artifacts phase-locked to heartbeat and auditory tone events were computed by epoching the data from -500 ms to 1000 ms around the ECG R-peak and tone onset, respectively, and averaging across epochs. Then, the template HEP and AEP were subtracted from the EEG signals at each instance of the ECG R-peak and tone onset, respectively. To

remove delta-band activity linked to ocular artifacts, a regression-based approach was employed[103]. First, a virtual electrooculography (EOG) channel was computed as the difference between signals from EEG electrodes Fp1 and Fp2. Then, linear regression was used to remove the resulting EOG signal from the EEG signals. After removal of evoked responses and artifacts from the broadband signal, frontal delta synchrony was recomputed (Fig. S1) exactly as before (Fig. 2A).

## Statistics and reproducibility
To identify the network of delta oscillations whose synchrony differed across correct and incorrect responses in the heartbeat detection task, network-based statistics were employed[104]. First, a threshold was computed as the 95th percentile of all entries in the connectivity (PLI) matrix averaged across trials and participants. Second, the largest connected component (network) was identified from the averaged and thresholded PLI connectivity matrix. To assess the statistical significance of synchrony in this network, the PLI was averaged within the network for each participant and condition (correct or incorrect responses in the heartbeat detection task). From this, a t-statistic was computed for the network. A permutation test was then used to compute a p-value for the network by shuffling the condition labels across trials within each participant and recomputing the t-statistic 10,000 times. To identify clusters across time and space where HEP amplitude correlated with delta phase synchrony in the identified network, a cluster-based permutation test was used[105]. To obtain a test statistic, Pearson's r was computed across participants between PLI in the network and HEP amplitudes at each point in time and space. In each permutation, PLI in the network was shuffled across participants. To assess the opposition of the phase difference between AM-tACS and frontal delta oscillations across early and late AM-tACS conditions (Fig. S3), we employed the phase opposition sum[106]. To assess phase-dependent modulation of frontal delta synchrony by AM-tACS (Fig. S4), we fit a sine function to the single-trial data using non-linear least-squares optimization. To assess the significance of the phase opposition sum or amplitude of phase-dependent modulation within each participant, we employed a permutation test. The labels (condition or AM-tACS-delta phase difference) were permuted across trials, and the test statistic was recomputed 10,000 times.

In total, 25 participants were assessed as part of this repeated-measures study (one participant was excluded due to ceiling performance, see methods section "Participants"). We did not have robust assumptions about the relevant parameters (i.e., effect size) before collecting the data, severely limiting the conclusions about Type I/II errors that can be drawn from such an analysis. Thus, the sample size was chosen to match that of a prior study on the effects of AM-tACS on perception[36]. In the absence of AM-tACS, each participant performed 60 trials of each of the heartbeat and tone detection tasks. In the presence of AM-tACS, each participant performed 120 trials of each of the heartbeat and tone detection tasks. For each task, an equal number of trials were performed per AM-tACS and sound delay condition (see Fig. 1).

## Reporting summary
Further information on research design is available in the Nature Portfolio Reporting Summary linked to this article.

## Data availability
The summary-level data generated in this study are provided in the supplementary data file. The raw EEG and behavioral data underlying this study are publicly available on G-Node (https://gin.g-node.org/davidhaslacher/commsbio-heartbeat-perception).

## Code availability
Python software using the publicly available MNE-Python (https://github.com/mne-tools/mne-python) and CLAM-NIBS packages was used to evaluate the data. A previously published algorithm (SASS) was used to assess EEG in the presence of electric stimulation artifacts (https://github.com/davidhaslacher/sass). Any scripts will be made available upon request.

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

## Acknowledgements

This project was supported by the Max Planck Institute for Human Cognitive and Brain Sciences through the MBB Young Scientist Award given to David Haslacher. This work was also supported in part by the European Research Council (ERC) under the project NGBMI (759370) and TIMS (101081905), and the Einstein Stiftung Berlin.

## Author contributions

D.H. and P.R. designed the study, collected the data, analyzed the data, and wrote the manuscript. A.C., A.R., E.P., A.B. and N.R.S. wrote the manuscript. V.N. and A.V. designed the study and wrote the manuscript. S.R.S. designed the study, acquired funding, provided laboratory resources, and wrote the manuscript.

## Funding

## Competing interests

The authors declare no competing interests.

## Additional information

**Supplementary information** The online version contains
supplementary material available at

Surjo R. Soekadar.

**Peer review information** *Communications Biology* thanks Florian H.
Kasten and Nico Adelhöfer for their contribution to the peer review of this
work. Primary Handling Editors: Christian Beste and Jasmine Pan. A peer
review file is available.

**Publisher's note** Springer Nature remains neutral with regard to
jurisdictional claims in published maps and institutional affiliations.

