## [Transparent Peer Review file · Communications Biology]

Heartbeat perception is causally linked to frontal delta oscillations

Corresponding Author: Professor Surjo Soekadar

Version 0:

Reviewer comments:

Reviewer #1

(Remarks to the Author)
Review

Haslacher et al. present a new study investigating the causal role of delta oscillations for interoception in a heart-beat detection task using amplitude modulated tACS. They find a modulation of detection performance depending on AM-tACS phase, as well as a negative correlation between performance and frontal delta synchrony.

I reviewed an earlier version of this manuscript for another journal. The authors did not address most of my previous comments. So, I will repeat them here with a short note explaining if and how the comment was addressed. In its current form I cannot recommend the paper for publication.

1) The authors use amplitude modulated tACS (AM-tACS), which is a novel form of tACS that many readers may not be familiar with, without providing much background on the principles of the method (and its' differences from conventional tACS). They should motivate more clearly why this technique was used and the methodological challenges that come with it. Generally, the method is supposed to avoid stimulation artifacts in the lower frequencies which are of interest for EEG analysis. However, it has been shown that AM-tACS is not completely artifact free at those frequencies due to non-linear properties of stimulation and recording electronics (Kasten et al., 2018). It should also be noted that most of the evidence for its efficacy comes from the authors' lab and other works have indicated that higher stimulation intensities might be necessary to achieve stimulation effects comparable to those of conventional tACS using pure sine waves (Negahbani et al., 2018). While the authors address these issues by using additional artifact cleaning techniques (SASS), and a stimulation intensity almost 10-fold higher than conventional tACS (+/- 10 mA), they do not explain any of these decisions to the readers. In particular, it should be made clear that the method is much less established than conventional tACS.

Note: The authors have added an extensive discussion on the limitations of their approach, which makes it much more transparent for the average reader. This comment has been resolved.

2) Performance of AM-tACS artifact cleaning is not transparently reported in the paper or the supplements. The authors only depict comparisons of PLI topographies. However, it is usually most informative to compare power spectra (and their topographies) before and after artifact cleaning to assess artifact suppression performance. Phase coupling measures are particularly vulnerable to spurious stimulation artifacts and even though topographies appear plausible, any residual artifact may bias comparisons if not accounted for. It is therefore crucial that the authors provide strong evidence for the success of their artifact removal, and that any residual artifacts cannot explain the current results. Currently it remains unclear if EEG results obtained during stimulation may be driven by insufficient artifact suppression.

Note: This comment has not been addressed by the authors in the new version of the manuscript.

3) The authors apply AM-tACS with a carrier frequency of 5 kHz, but record EEG at only 2 kHz sampling rate. I would expect that this may cause substantial aliasing artifacts in the signal that could affect lower frequencies in the EEG (and other electrophysiological measures).

Note: This comment has not been addressed by the authors.

4) The authors use a conventional 2-electrode setup using small circular electrodes. It remains relatively unclear which part of the frontal cortex is being stimulated. The topography of EEG delta synchrony seems to indicate the strongest frontal synchrony in the anterior PFC, while with the electrode montage I would expect that the electric field stimulates more medial/posterior regions of the frontal cortex. Which may miss the regions of interest seen in the EEG. I would suggest to add a simulation of the electric field to indicate the stimulated brain regions more clearly (e.g. using ROAST or simnibs (Huang et al., 2019; Thielscher et al., 2015)). A source localization of the delta synchrony effect may also be beneficial. With 64 EEG channels such a localization should be possible, even though the precision may not be optimal.

Note: This has been partially resolved. The authors added an electric field simulation to the supplementary materials. However, there is no comparison to the source of delta oscillations. So it is not entirely clear if the field distribution offers a good overlap with the delta activity.

5) Assessing a modulation of Delta PLI by Delta AM-tACS phase does not seem appropriate. A change in delta PLI with AM-tACS phase at the same frequency would imply that synchrony in the delta frequency range changes at a rate much faster than a single cycle of the oscillation itself. This seems conceptually problematic. I am wondering if the phasic modulation of delta PLI reported in the paper is simply an artifact resulting from an oversampling of the synchronization measure in the analysis.

Note: This point has not been addressed.

6) Measures of synchronization like the PLI can be biased by volume conduction in the EEG. The authors do not explain how this was addressed in the current study.

Note: This point has not been addressed.

7) The two main findings of the study are that higher delta synchrony leads to lower performance in heart-beat detection, while synchronizing delta oscillations with AM-tACS leads to a phase dependent modulation of heart-beat detection performance, indicating that delta activity synchronized with the heard beat may be beneficial for interoception. These two findings seem in contradiction, yet this is never discussed in the paper. In general, the authors are not very clear about the direction of the effect they expected in their work and how they relate to previous findings.

Note: This has not been addressed.

8) It is not reported which modulation frequency in the delta range was applied for AM-tACS. Was Delta frequency individually chosen for each participant or was a fixed frequency used? It seems the frequency may have been controlled by the real-time machine, similar to the phase. But the implementation is not disclosed.

Note: This has not been addressed.

Minor

AM-tACS was applied at individual intensities up to 10 mA. What was the actual stimulation intensity across the sample?

Note: This has been addressed.

An additional minor comment while reading the current version of the manuscript:

Ln 212 – 219 seem redundant with Ln 221 following...

References

- Huang, Y., Datta, A., Bikson, M., & Parra, L. C. (2019). Realistic volumetric-approach to simulate transcranial electric stimulation—ROAST—a fully automated open-source pipeline. *Journal of Neural Engineering*, 16(5), Article 5. <https://doi.org/10.1088/1741-2552/ab208d>
- Kasten, F. H., Negahbani, E., Fröhlich, F., & Herrmann, C. S. (2018). Non-linear transfer characteristics of stimulation and recording hardware account for spurious low-frequency artifacts during amplitude modulated transcranial alternating current stimulation (AM-tACS). *NeuroImage*, 179, 134–143. <https://doi.org/10.1016/j.neuroimage.2018.05.068>
- Negahbani, E., Kasten, F. H., Herrmann, C. S., & Fröhlich, F. (2018). Targeting alpha-band oscillations in a cortical model with amplitude-modulated high-frequency transcranial electric stimulation. *NeuroImage*, 173, 3–12. <https://doi.org/10.1016/j.neuroimage.2018.02.005>
- Thielscher, A., Antunes, A., & Saturnino, G. B. (2015). Field modeling for transcranial magnetic stimulation: A useful tool to understand the effects of TMS? 2015 37th Annual International Conference of the IEEE Engineering in Medicine and Biology Society (EMBC), 222–225. <https://doi.org/10.1109/EMBC.2015.7318340>

Reviewer #2

(Remarks to the Author)

This manuscript tackles an ambitious and timely question: can closed-loop, ECG-triggered delta-frequency AM-tACS over frontal cortex causally modulate the neural and behavioural indices of heartbeat perception? The real-time stimulation–recording pipeline, extensive EEG cleaning procedures, and the effort to link predictive coding with interoception are clear strengths that will interest Communications Biology’s readership. My main reservations, however, concern (i) how convincingly the chosen heartbeat-discrimination task and statistics support the conclusions, (ii) whether artefacts or peripheral co-stimulation could explain the reported EEG and behavioural effects, and (iii) the degree to which the paper’s theoretical claims go beyond what the data currently justify. I outline these issues below.

Major points:

- Recent work has highlighted conceptual and psychometric limitations of heartbeat-detection paradigms. Please provide full signal-detection metrics (d' and criterion) in addition to confidence-weighted accuracy, and discuss critical papers such as Desmedt et al. 2022. With regards to associated statistics, several key effects (e.g. PLI difference, $t(23)=1.61$, $p = 0.044$ one-tailed) are only marginally significant and uncorrected for multiple testing. I strongly recommend controlling the false-discovery rate, reporting effect sizes with confidence intervals, and including a priori power calculations so that readers can judge evidential strength.
- Stimulation at 6.28 ± 0.97 mA pk-pk is far above typical tACS intensities and may activate retina or cutaneous nerves even below subjective threshold. Although SASS was applied, the paper shows no sham-tACS or ICA demonstration that residual ECG or tACS artefacts are negligible. Moreover, without an active or site-control condition and without post-session sensory questionnaires, sensory co-stimulation cannot be ruled out. Please provide explicit artefact-suppression validation (before/after plots, ideally sham data), document formal blinding and sensory-threshold checks, and expand the limitations section if new control data cannot be gathered.
- The introduction states that frontal delta oscillations “suppress HEPs” yet stops short of explaining how delta-phase synchrony implements interoceptive predictions. Please flesh out this mechanism (e.g. through predictive-coding language of precision weighting) and restructure the Discussion so that each key result is interpreted in that framework rather than devoting disproportionate space to general AM-tACS caveats.

Minor points:

- abstract: moderate "plays a critical role".
- introduction: justify 250 ms and 750 ms tone delays given pulse-transit variability, and explain the choice of a 785 Hz deviant.
- results: plot paired within-subject data with connecting lines in Figures 2B and 3A to visualize individual effects clearly.
- discussion: merge the duplicate emotion-theory paragraphs.

Version 1:

Reviewer comments:

Reviewer #1

(Remarks to the Author)

The authors have adequately addressed my concerns and I support publication of the manuscript.

Reviewer #2

(Remarks to the Author)

The authors have satisfactorily addressed all my previous concerns. I have no further comments, and I recommend the manuscript for publication.

Point-by-point response to the reviewers

We thank the reviewers for their insightful, concise, and instrumental feedback on our manuscript. We have incorporated all suggested changes and believe that the impact and clarity of our work has substantially improved thanks to the reviewer's valuable feedback. Please find below a point-by-point response to the suggestions. In the revised manuscript, all changes are marked in red.

Reviewer 1

2) Performance of AM-tACS artifact cleaning is not transparently reported in the paper or the supplements. The authors only depict comparisons of PLI topographies. However, it is usually most informative to compare power spectra (and their topographies) before and after artifact cleaning to assess artifact suppression performance. Phase coupling measures are particularly vulnerable to spurious stimulation artifacts and even though topographies appear plausible, any residual artifact may bias comparisons if not accounted for. It is therefore crucial that the authors provide strong evidence for the success of their artifact removal, and that any residual artifacts cannot explain the current results. Currently it remains unclear if EEG results obtained during stimulation may be driven by insufficient artifact suppression.

Performance of AM-tACS artifact cleaning is now more thoroughly reported, as requested by the reviewer. In addition to PLI topographies (Fig. S2), we have now added power spectra (Fig. S6A) and delta power topographies (Fig. S6B) before and after artifact cleaning. We have adapted the corresponding sentence in the main text to reflect these new results (lines 287 – 290): “To mitigate this possibility, we have applied SASS, an artifact suppression technique specifically designed for use with AM-tACS (67), and confirmed that frontal delta synchrony after application of SASS is comparable to that in absence of AM-tACS (Fig. S2). We also provide a similar analysis for the EEG power spectral density and delta power (Fig. S6).”

3) The authors apply AM-tACS with a carrier frequency of 5 kHz, but record EEG at only 2 kHz sampling rate. I would expect that this may cause substantial aliasing artifacts in the signal that could affect lower frequencies in the EEG (and other electrophysiological measures).

EEG is digitally sampled after an antialiasing hardware (analog) filter, typically at one quarter of the sampling rate (125 Hz here). Thus, attenuation of our 8 kHz carrier frequency due to this antialiasing filter is extremely strong, leading to practical elimination of that signal before it is digitally sampled. Low-frequency artifacts do exist (Fig. S6A) but originate from non-linearities in the stimulation and recording hardware [1].

4) The authors use a conventional 2-electrode setup using small circular electrodes. It remains relatively unclear which part of the frontal cortex is being stimulated. The topography of EEG delta synchrony seems to indicate the strongest frontal synchrony in the anterior PFC, while with the electrode montage I would expect that the electric field stimulates more medial/posterior regions of the frontal cortex. Which may miss the regions of interest seen in the EEG. I would suggest to add a simulation of the electric field to indicate the stimulated brain regions more clearly (e.g. using ROAST or simnibs (Huang et al., 2019; Thielscher et al., 2015). A source localization of the delta synchrony effect may also be beneficial. With 64 EEG channels such a localization should be possible, even though the precision may not be optimal.

Unfortunately, source reconstructions were of insufficient quality to precisely localize the origin of delta oscillations. As multiple EEG electrodes near the tACS electrodes needed to be excluded from the analysis, the scalp was not homogeneously sampled by the EEG electrodes and a spatial bias in the reconstructed source activity could be observed. Furthermore, we did not digitize electrode locations or perform individual anatomical brain scans for accurate forward modeling. Nevertheless, sensor-space topographies of delta PLI (Fig. S2) and power (Fig. S6B) show that the location of delta oscillations overlaps with the location of the AM-tACS electric field (Fig. S5).

5) Assessing a modulation of Delta PLI by Delta AM-tACS phase does not seem appropriate. A change in delta PLI with AM-tACS phase at the same frequency would imply that synchrony in the delta frequency range changes at a rate much faster than a single cycle of the oscillation itself. This seems conceptually problematic. I am wondering if the phasic modulation of delta PLI reported in the paper is simply an artifact resulting from an oversampling of the synchronization measure in the analysis.

By design of our experiment (Fig. 1), the phase difference between AM-tACS and delta oscillations remained approximately constant throughout the duration of each trial (approximately 10 seconds). Thus, synchrony in the delta frequency range does not change faster than a single cycle of the oscillation itself but is assumed to remain approximately constant throughout each 10 second trial. Thus, each data point in Fig. S4 represents a single trial.

6) Measures of synchronization like the PLI can be biased by volume conduction in the EEG. The authors do not explain how this was addressed in the current study.

The PLI was chosen precisely because it mitigates the effect of volume conduction. By ignoring 0° and 180° phase differences between two sources, any interference due to volume conduction that would bias conventional measures of synchronization (like the phase locking value, PLV) is suppressed [2]. This makes our choice of synchronization measure robust to electric stimulation artifacts and volume conduction of neurogenic sources. This is now explained in the methods section on EEG processing (lines 444 – 448): “The PLI was chosen because it mitigates the effect of volume conduction. By ignoring 0° and 180° phase differences between two sources, any interference due to volume conduction that would bias conventional measures of synchronization (like the phase locking value, PLV) is suppressed. This makes the PLI measure robust to electric stimulation artifacts and volume conduction of neurogenic sources”.

7) The two main findings of the study are that higher delta synchrony leads to lower performance in heart-beat detection, while synchronizing delta oscillations with AM-tACS leads to a phase dependent modulation of heart-beat detection performance, indicating that delta activity synchronized with the heard beat may be beneficial for interoception. These two findings seem in contradiction, yet this is never discussed in the paper. In general, the authors are not very clear about the direction of the effect they expected in their work and how they relate to previous findings.

We now clarify this issue in the discussion (lines 245 – 251): “An important methodological difference between our study and prior work (55) investigating the role of brain oscillations in mediating the perception of rhythmic sensory input should also be emphasized. We did not investigate an alignment (synchronization) of delta oscillations with the heartbeat, which was difficult in the presence of cardiac-related artifacts in the EEG. To mitigate this issue, we

investigated synchronization of delta oscillations within a prefrontal brain network using the PLI, a measure of synchronization robust to volume conduction originating from sources such as the cardiac artifact.” Our hypotheses are clearly stated in the introduction (lines 63 – 64): “we hypothesized that phase synchrony of FDOs is causally and negatively linked to heartbeat-evoked potentials and heartbeat detection”. This emerged from prior (unpublished) pilot data in our lab, which had already suggested the same effect.

8) It is not reported which modulation frequency in the delta range was applied for AM-tACS. Was Delta frequency individually chosen for each participant or was a fixed frequency used? It seems the frequency may have been controlled by the real-time machine, similar to the phase. But the implementation is not disclosed.

As can be seen in Fig. 1D, AM-tACS was adapted to the heartbeat, such that the peak of AM-tACS was either early (+250 ms) or late (+750 ms) relative to the ECG R-peak. Both frequency and phase of the AM-tACS envelope were continually adjusted to the individual heartbeat. After each ECG R-peak, the frequency was adjusted to match the average heart rate over the last 10 heartbeats, while the phase was adjusted to achieve the desired early or late timing. This is now mentioned in the methods section on transcranial alternating current stimulation (lines 388 – 391): “Both frequency and phase of the AM-tACS envelope were continually adjusted to the individual heartbeat. After each ECG R-peak, the frequency was adjusted to match the average heart rate over the last 10 heartbeats, while the phase was adjusted to achieve the desired early or late timing”.

Minor

An additional minor comment while reading the current version of the manuscript:

Ln 212 – 219 seem redundant with Ln 221 following...

This has been resolved.

Reviewer 2

Recent work has highlighted conceptual and psychometric limitations of heartbeat-detection paradigms. Please provide full signal-detection metrics (d' and criterion) in addition to confidence-weighted accuracy, and discuss critical papers such as Desmedt et al. 2022. With regards to associated statistics, several key effects (e.g. PLI

difference, $t(23)=1.61$, $p = 0.044$ one-tailed) are only marginally significant and uncorrected for multiple testing. I strongly recommend controlling the false-discovery rate, reporting effect sizes with confidence intervals, and including a priori power calculations so that readers can judge evidential strength.

We now assess full signal-detection metrics (d' and criterion) wherever we assess confidence-weighted accuracy in the results section. We also discuss conceptual and psychometric issues of heartbeat detection paradigms in a new discussion section titled “Limitations of heartbeat detection paradigm” (lines 332 – 353). Regarding the statistical evaluation of PLI difference (Fig. 2B), correction for multiple comparisons across space are naturally handled by the cluster (network) based permutation test that results in the depicted effect. In other words, the statistics reported in Fig. 2B are simply those for the significant cluster (network) depicted in Fig. 2A that emerges from the test. We also assess whether PLI in this network is different across tasks (Fig. 2B), but this does not require a correction for multiple comparisons, as it tests a different hypothesis (we do not expect or find an effect here). As requested by the reviewer, we now also report confidence intervals for all effect sizes in the results section. We do not report a priori power calculations, however, as we did not have robust assumptions about the relevant parameters before collecting the data, severely limiting the conclusions about type I/II errors that can be drawn from such an analysis.

Stimulation at 6.28 ± 0.97 mA pk-pk is far above typical tACS intensities and may activate retina or cutaneous nerves even below subjective threshold. Although SASS was applied, the paper shows no sham-tACS or ICA demonstration that residual ECG or tACS artefacts are negligible. Moreover, without an active or site-control condition and without post-session sensory questionnaires, sensory co-stimulation cannot be ruled out. Please provide explicit artefact-suppression validation (before/after plots, ideally sham data), document formal blinding and sensory-threshold checks, and expand the limitations section if new control data cannot be gathered.

We now provide more extensive evidence of artifact cleaning efficacy by assessing PLI topographies (Fig. S2), power spectra (Fig. S6A), and topographies of delta power (Fig. S6B) in data recorded without AM-tACS, with AM-tACS before SASS, and with AM-tACS after SASS. We also provide evidence that ECG artifacts are negligible in Fig. S1. While we are unable to rule out contributions of sensory stimulation effects and have not collected post-session sensory

questionnaires, these limitations are described in the discussion section “Limitations of transcranial alternating current stimulation approach” (lines 321 – 330).

The introduction states that frontal delta oscillations “suppress HEPs” yet stops short of explaining how delta-phase synchrony implements interoceptive predictions. Please flesh out this mechanism (e.g. through predictive-coding language of precision weighting) and restructure the Discussion so that each key result is interpreted in that framework rather than devoting disproportionate space to general AM-tACS caveats.

We have restructured the discussion to interpret our key results in terms of the predictive coding framework. This is now summarized in a separate discussion section: “Interoceptive predictive coding” (lines 217 – 251). However, it was necessary to maintain a lengthy discussion of AM-tACS caveats, as several prior reviewers insisted on the importance of these limitations.

Minor points:

abstract: moderate “plays a critical role”.

Done.

introduction: justify 250 ms and 750 ms tone delays given pulse-transit variability, and explain the choice of a 785 Hz deviant.

This is now explained in the methods section (lines 401 – 406): “The timing of early and late tones was chosen to match that of a prior study implementing the same paradigm, but with pulse measurements from the finger (11). In that study, early tones were played immediately after the pulse wave, whereas late tones were played ~500 ms after the pulse wave. Our timing was subsequently chosen based on evidence that the pulse wave arrives at the finger ~250 ms after the ECG R-peak (96). Tone frequency was also chosen to match the same prior study: normal tones had a frequency of 800 Hz, whereas deviant tones had a frequency of 785 Hz”.

results: plot paired within-subject data with connecting lines in Figures 2B and 3A to visualize individual effects clearly.

We sincerely thank the reviewer for this thoughtful suggestion. After careful consideration, we decided not to implement this change, as we felt it might unintentionally give the impression that the contrast between the two stripplots in Fig. 2B represents the main effect of interest.

Our intention is to emphasize the deviation from zero in the experimental condition (left panel), with the control condition (right panel) serving as a contextual reference. This is reflected in the p-values shown above each stripplot. Moreover, we found that adding connecting lines increased visual complexity and reduced the overall clarity of the figure.

discussion: merge the duplicate emotion-theory paragraphs.

Done.

We thank the reviewers again for their valuable and constructive feedback that helped us to improve the manuscript in clarity and impact.

References

- [1] Kasten, Florian H., et al. "Non-linear transfer characteristics of stimulation and recording hardware account for spurious low-frequency artifacts during amplitude modulated transcranial alternating current stimulation (AM-tACS)." *Neuroimage* 179 (2018): 134-143.
- [2] Stam, Cornelis J., Guido Nolte, and Andreas Daffertshofer. "Phase lag index: assessment of functional connectivity from multi channel EEG and MEG with diminished bias from common sources." *Human brain mapping* 28.11 (2007): 1178-1193.